# Impact on Myanmar's Logistics Flow of the East–West and Southern Corridor Development of the Greater Mekong Subregion—A Global Logistics Intermodal Network Simulation

**Takuya Yamaguchi [1], Ryuichi Shibasaki [2,*], Hiroyuki Samizo [3] and Hisanari Ushirooka [3]**

1    Department of Systems Innovation, School of Engineering, The University of Tokyo, Tokyo 113-8656, Japan;
     takuya.gs.0627@gmail.com
2    Resilience Engineering Research Center/Department of Technology Management for Innovation,
     School of Engineering, The University of Tokyo, Tokyo 113-8656, Japan
3    Nippon Koei Co., Ltd., Tokyo 102-8539, Japan; a5247@n-koei.co.jp (H.S.); a5536@n-koei.co.jp (H.U.)
*    Correspondence: shibasaki@tmi.t.u-tokyo.ac.jp; Tel.: +81-3-5841-6546

**Abstract:** This study focuses on container shipping in Myanmar, which is expected to grow manifold in the near future, given the country's rapid economic growth rates. This study simulates the impact of Myanmar's logistics policies on container shipping. These initiatives include the improvement of the East–West Corridor of the Greater Mekong Subregion and the development of the Southern Corridor and Dawei port. The global logistics intermodal network simulation model including both maritime shipping and land transport, is applied to the land-based southeast Asia (ASEAN) region. The estimated results obtained for several different scenarios are crosschecked and compared with the available information on observed flows. Based on the simulation results, the authors conclude that policies that reduce cross-border barriers and improve service levels in Dawei port would result in Thailand using Myanmar's ports for their cargo as well.

**Keywords:** global logistics simulation; intermodal freight transport network; economic corridor; Myanmar; terrestrial ASEAN; Greater Mekong Subregion; East–West Corridor; Southern Corridor; Dawei port; GLINS model

## 1. Introduction

In recent years, with the growth of the world economy, globalization, and the subsequent expansion of trade, the importance of international maritime container transport is increasing. Whereas developed countries remain the nucleus of global trade, emerging and developing countries are rapidly increasing their participation in international trade, which is simultaneously a reason and a result of the remarkable economic growth achieved by these countries. The ASEAN region is currently the focus of attention, not only because of its high economic growth, but also because of its proximity to China, which is promoting the Belt and Road Initiative.

Myanmar is considered the last new frontier in Southeast Asia, with a GDP growth rate of 6–8% since 2011 [1]. However, most of Myanmar's logistics infrastructure was developed during the British colonial era, and is in urgent need of upgradation and renewal. In other words, significant growth of investment in Myanmar's logistics infrastructure is required and expected in the future. Hence, in this context, for ensuring efficient use of limited resources to improve the national economy, it is significant to propose the best scenario based on quantitative policy simulations on logistics infrastructure. Further, formulating a sustainable infrastructure policy is currently important, not only from an economic point of view but also from the environmental point of view. In this respect, the intermodal simulations in this study will contribute to a quantitative discussion on the environmental impact of different modes of transport, based on their characteristics.

## 2. Literature Review

As summarized by Shibasaki [2], several studies, such as Tavasszy et al. [3], based on a path size logit model and ITF-OECD [4], based on a shortest path search model, developed a global intermodal logistics simulation model other than those developed by the authors (which will be explained later). As discussed in Holguín-Veras et al. [5], in studies on large-scale logistics simulations including transport mode choice, it is generally difficult to develop a model to contain various elements similar to those in supply chain models because the data is unavailable, thus, a simpler model tends to be used. Even if such simulation models are applied to developing countries, obtaining data is much more difficult; additionally, the capacity constraint of infrastructure is more serious in developing countries due to the insufficient infrastructure, although the traffic growth rate is much faster there. Recently, several studies conducted logistics network simulation for emerging and developing countries, such as Aritua et al. [6], focusing on South Africa and India using a gravity model, Meersman et al. [7], comparing generalized chain cost (which was defined in Hassel et al. [8]) of each route in the Eurasian continent in the context of China's Belt and Road Initiative, Verhaeghe et al. [9] developing the network optimization model by combining the path size logit model [3] with a genetic algorithm and applying to Indonesia, and Kawasaki et al. [10] and Shibasaki and Kawasaki [11] applying the same concept model [2,12,13] as this study to the African continent and the South Asian region, respectively.

Among them, Table 1 summarizes related literature for quantitative policy simulations on international logistics infrastructure in the ASEAN region and Myanmar. Several studies have implemented an international freight simulation model using the similar model in this paper, including Shibasaki et al. [14], Iwata et al. [15] and Kosuge et al. [16]. Shibasaki et al. [17] developed an international logistics simulation model for the ASEAN region and analyzed the impact of a batch of logistics policies on the entire ASEAN region, but did not focus on specific policies in each country such as Myanmar. Iwata et al. [15] focused on Lao PDR and the surrounding countries and used a logistics simulation model to evaluate land transport development and port development, but focused specifically on Laos, which is a landlocked country, and did not focus on Myanmar. Kosuge et al. [16] conducted a logistics simulation for the future of Cambodia, but the other regions of land-based ASEAN countries (hereafter referred to as 'terrestrial ASEAN'), which consist of Cambodia, Lao PDR, Thailand, Vietnam and Myanmar were simplified and not focused upon. Another similar simulation model was developed by Kawasaki et al. [17], an inland cargo flow model that takes into account the additional costs caused by the variability of shipment time at the border and ports. They analyzed five scenarios for cross-border transport between Laos and ports in Thailand and Vietnam to evaluate the effect of improving the reliability of the border and ports, but they did not focus on Myanmar.

Several studies have focused on the Greater Mekong Subregion (GMS) in particular. Kawasaki et al. [18] used data on the preference of shippers engaged in cross-border transport in the GMS to estimate the value of shipping time variability, but this had not been linked to individual country policy analysis. Further, Strutt et al. [19] used a database to simulate trade facilitation and analyzed the GMS development policies; Stone and Strutt [20] examined multiple scenarios on the potential for GDP growth in the GMS; and Tansakul et al. [21] focused on the East–West Corridor (EWC) in the GMS and applied the Analytical Hierarchy Process to examine the effects of various factors enhanced by the trade facilitation. However, scopes of these researches were not based on a logistics network specific.

**Table 1.** Summary of literature review on quantitative policy simulations on international logistics in the terrestrial ASEAN region and Myanmar.

| Papers | Developing a Simulation Model Considering Both Maritime and Land Transport Network | Including the Entire Terrestrial ASEAN | Analyzing Myanmar's Policy |
|---|---|---|---|
| Shibasaki et al. [14] | x | x | |
| Iwata et al. [15] | x | x | |
| Kosuge et al. [16] | x | x | |
| Kawasaki et al. [17] | x | x | |
| Kawasaki et al. [18] | x | | |
| Strutt et al. [19] | | | |
| Stone and Strutt [20] | | x | x |
| Tansakul et al. [21] | | x | |
| Kudo and Kumagai [22] | | | x |
| Black and Kyu [23] | | | x |
| Zin [24] | | | x |
| Nam and Win [25] | | | x |
| Sukdanont et al. [26] | x * | | x |
| Isono and Kumagai [27] | x ** | x | x |
| Isono [28] | | x | |
| Shepherd and Wilson [29] | | x | |
| Sy et al. [30] | | x | |
| Opasanon and Kitthamkesorn [31] | | x | |
| Jiang et al. [32] | x | | |
| Suvabbaphakdy et al. [33] | | x | |
| Zheng et al. [34] | | x | |
| This study | x | x | x |

* Only coastal (domestic) maritime shipping is considered; ** includes both the maritime and land transport network in a simplified manner, but mainly focuses on economic impact.

Several simulation studies focused on Myanmar's logistics network and related policies. Kudo and Kumagai [22] used a general equilibrium geographic model to simulate a bipolar economic system with Yangon and Mandalay and compared the results among the different GRDP growth scenarios in Myanmar, but the simulation was not based on a logistics network and the area covered was only within Myanmar. Black and Kyu [23] analyzed Myanmar's imports and exports with a focus on Mandalay's dry ports, but did not consider Myanmar's trade relations with other countries, such as its relationship with ASEAN on land. Zin [24] also focused on Myanmar's dry ports, but did not consider their relationship with neighboring countries. Nam and Win [25] focused on Myanmar's intermodal system with a focus on inland waterway transport, but their interest was also limited to domestic Myanmar. Sukdanont et al. [26] conducted a route specific cost analysis of coastal and road intermodal transport in the region, but only analyzed freight transport in some specific routes between Thailand and Myanmar. Isono and Kumagai [27] simulated the development of Dawei port using the Geographical Simulation Model (IDE-GSM) on the global intermodal transport network including both maritime and land transport; however, their focus was on estimating the economic impact of the port on the surrounding areas and the simulation was not based on a detailed logistics network. Isono [28] similarly applied the IDE-GSM to estimate the economic effects of the infrastructure projects in Thailand, including the Southern Corridor (SC) in the GSM, but the simulations were not based on a detailed logistics network and did not focus on Myanmar.

Regarding other logistics simulations for the ASEAN region from the different viewpoints, Shepherd and Wilson [29] developed a gravity model to analyze the correlation be-

tween trade facilitation and various indicators in the ASEAN region. Similarly, Sy et al. [30] used a panel data to build an extended gravity model for the ASEAN region and analyzed the correlations between logistics performance and trade value, but none of them were based on a detailed logistics network and there were no policy analyses. Opasanon and Kitthamkesorn [31] developed a linear regression model and conducted a simulation case study of Thailand's largest customs, but the analysis was limited to the customs rather than a broader infrastructure policy. Moreover, some studies have focused on ASEAN's relationship with other regions. Jiang et al. [32] simulated the impact of the trade and multimodal transport corridors jointly constructed by the provinces of western China and the ASEAN countries on the neighboring countries, and calculated the choice behavior of freight transport using a logit model. However, the trade routes considered were limited and did not focus on Myanmar's infrastructure policy. Suvabbaphakdy et al. [33] simulated bilateral trade between 16 countries, including the ASEAN, but did not focus on individual countries and not use a detailed logistics network. Zheng et al. [34] developed and simulated a system dynamics model of regional economic development and air logistics interaction in Guangxi Zhuang Autonomous Region, but the emphasis was on China.

In summary, as shown in Table 1, there are no papers that satisfy all the following criteria: (1) developing a detailed logistics simulation model considering both maritime and land networks, (2) including the entire terrestrial ASEAN region and, (3) analyzing Myanmar's policy. Therefore, this study applies the existing network assignment model to simulate global maritime container shipping and land transport in terrestrial ASEAN region. Using the model, scenario simulations of current and future logistics infrastructure policies in Myanmar, which is one of the terrestrial ASEAN countries penetrated by several corridors of the GMS, are performed. The simulations also include the impact on the entire terrestrial ASEAN countries.

### 3. International Logistics Environment in Myanmar

Figure 1 shows a logistics network in Myanmar including the major nodes and corridors, which are described below.

Thilawa and Yangon ports are important centers of international logistics and gateways to international trade in Myanmar. These ports are located in or near Yangon, the largest city of Myanmar, which accounts for more than 10% of Myanmar's total population and about 25% of its GDP (Institute of Developing Economies). Most of the maritime containers in Myanmar are handled at either of these ports. As Yangon port is the older port and is narrow, it has limited scope for development to accommodate and meet the future, burgeoning demand for container handling. It cannot be maintained as the only gateway port of Yangon city, therefore, Thilawa port is being developed to accommodate the increased volumes of import and export cargo that are expected in conjunction with Myanmar's future development. Moreover, the surrounding area has been designated as a special economic zone, and many factories of foreign companies have expanded into the area and are expected to grow. In the rest of this study, Thilawa and Yangon port are collectively referred to as Thilawa port.

The EWC is one of the most important economic corridors in the GMS [36]. This corridor runs from east to west through Vietnam, Laos, Thailand and Myanmar. Focusing on the part in Myanmar, the main land transport route between Yangon and Bangkok (Thailand) overlaps the EWC from Yangon to Tak in Thailand, which is an important section from the perspective of Myanmar's international logistics environment. Trade between Myanmar and Thailand is currently conducted mainly by land, and this route is most commonly used. Although the Thai section of the EWC is well maintained, its Myanmar section is often flooded during the rainy season due to unpaved roads.

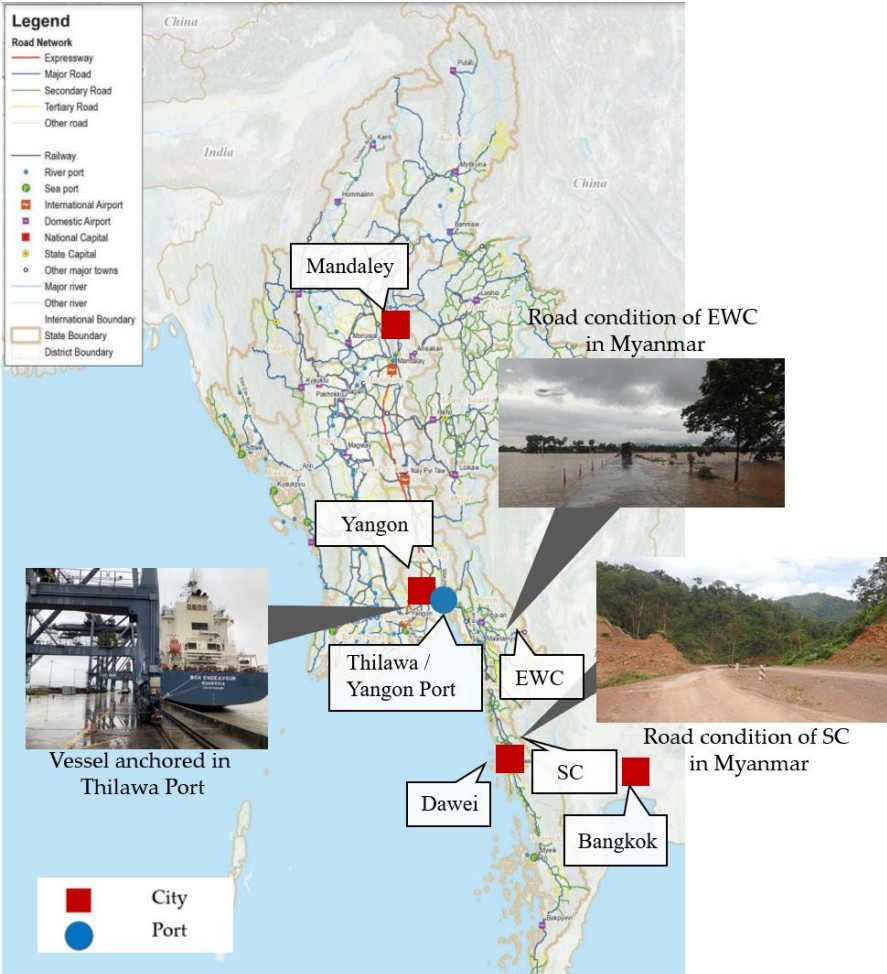

**Figure 1.** Logistics network in Myanmar including the major nodes and corridors. Source: Japan International Cooperation Agency (JICA) [35].

The SC, which is also part of the GMS economic corridor, runs from Ho Chi Minh City (Vietnam), through Phnom Penh (Cambodia) and Bangkok, to Dawei, which is a provincial city in southern Myanmar, about 600 km south of Yangon. Although the road between Dawei and Phu Nam Long on the Thai border has not yet been developed and this section of the road does not function as a corridor, the Thai stakeholders have positioned Dawei as an outer port of the Thai metropolitan area, for transport to India and Europe. Conversely, Myanmar's stakeholders are skeptical about the benefits of the port to Myanmar, as the Dawei–Bangkok route traverses through its territory; therefore, the priority of development is different between both countries. Such controversial projects should be carefully and quantitatively examined through the policy simulation model.

As mentioned above, there are many open issues regarding the development of a logistics infrastructure and its impacts in Myanmar; therefore, it would be useful to quantitatively verify each of them through simulation analysis.

## 4. Simulation Model

### 4.1. Overview of the Model

The global logistics intermodal network simulation (GLINS) model used in this study is based on the model developed by Shibasaki [2,12,13] and then applied in Shibasaki et al. [37]. The model also considers international land transport, in the sense that it does not use maritime shipping, not only maritime shipping and their hinterland transport. Figure 2 shows the structure of the model. The major difference between this study's model and the models used in Shibasaki et al. [14] and Iwata et al. [15] is that their models also

endogenized the decision on liner services by shipping companies; therefore, they had major challenges in practical aspects such as model fitness to the actual and policy scenario analysis, which made it difficult to simulate individual infrastructure policies. In this study's model, the level of liner services provided by shipping companies is exogenously given as a scenario, and the model is specialized for cargo assignment so that the model fitness and the accuracy of individual policy simulation can be significantly improved.

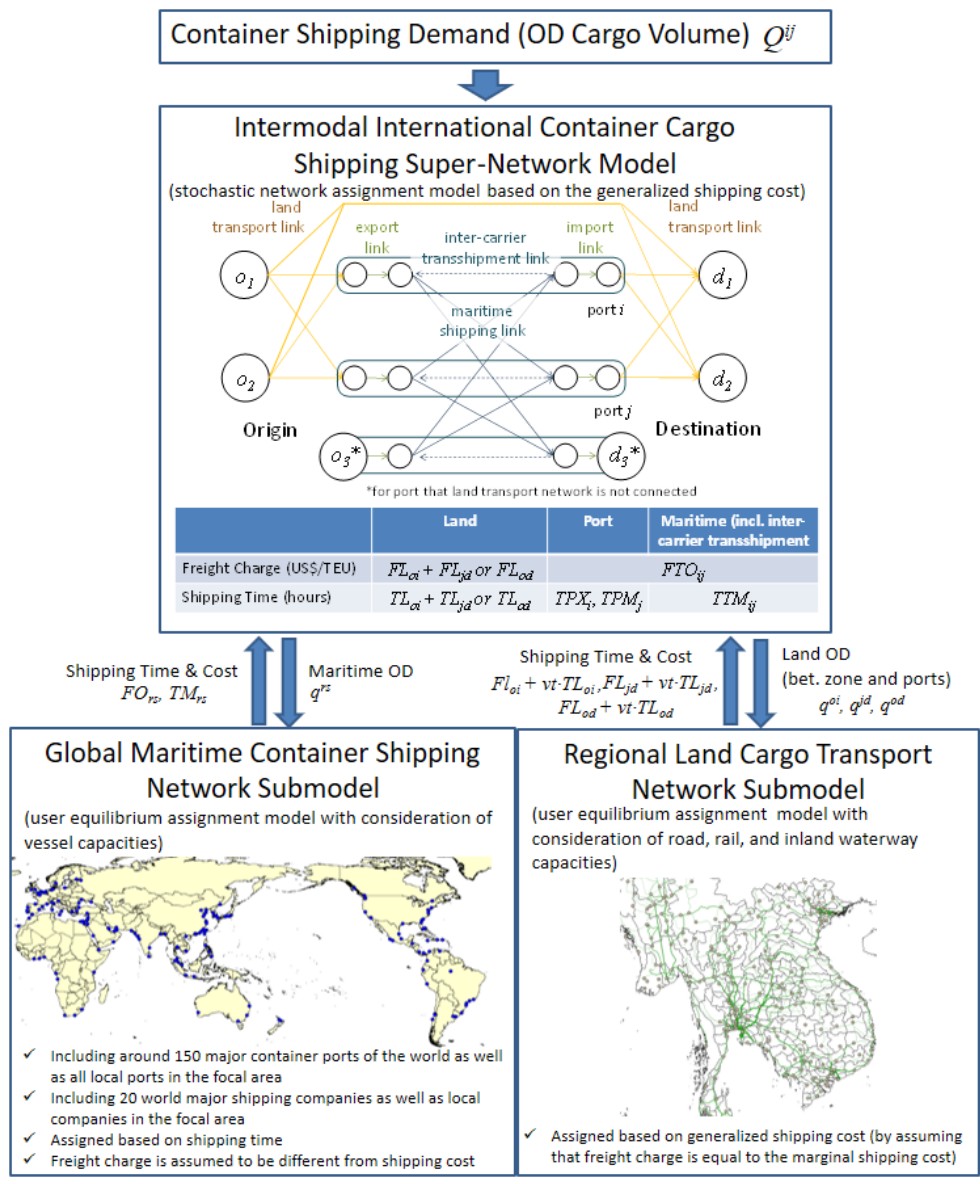

**Figure 2.** Structure of the GLINS (global logistics intermodal network simulation) model. Source: Modified Shibasaki [2].

The GLINS model is composed of two submodels, one based on the real network on the sea and the other based on land. There is one upper model on an intermodal virtual network that integrates them. In the upper model, a stochastic network assignment (Dial assignment) is used to allocate cargo to flow on other paths than those with the lowest link cost. In the assignment calculations for the submodels of maritime shipping and land transport networks, the user equilibrium assignment is applied to take the effects of congestion into account. As described in Section 2, consideration of the congestion effect is essential, especially for the simulations in developing countries, because capacity constraints of infrastructure are very critical there.

There are two main inputs in the GLINS model: network data, including distance, level of service and transport volume, for sea, port and land; and the interregional cargo shipping demand (OD matrix). The output is the container flow at each link, and by aggregating the output, the container handling volume at each port and the overall flow between ports can be calculated.

The GLINS model incorporates the cross-border coefficient $\lambda_a$, which is defined as the rate of the procedure cost and time of land-transit cargo to those of normal import/export cargo if crossing land national borders, as shown in Equation (1).

$$u'(x_a) = u(x_a) + \lambda_a(CBO_a + vt \cdot TBO_a) \tag{1}$$

where $x_a$ is a flow of link $a$, $u(x_a)$ is ordinary cost for a link (USD/TEU), $u'(x_a)$ is cost for a link that crosses the national boder, $CBO_a$ is addional monetary cost (USD/TEU) in border-crossing (which is set by country based on World Bank Group [38]), $TBO_a$ is addional time (USD/TEU) in border-crossing (same as above) and $vt$ is shipper's time value of freight (USD/TEU/hour). As stated in Section 2, the quantitative data for the simulations, including other parameters in all cost functions, is generally difficult to obtain especially for developing countries. Therefore, in the model of Shibasaki [2,12,13], they are often approximated by the interview survey results with stakeholders and alternative indicators are used to supplement the data.

*4.2. Input Data*

Based on Shibasaki et al. [14] (which is a previous study on logistics model simulation for Southeast Asia), the interregional shipping demand of cargo and maritime and land transport networks in 2016 is generated.

For a detailed analysis of the terrestrial ASEAN network, we added the ports of Da Nang and Khu Inong in Vietnam, Sihanoukville in Cambodia and Songkhla in Thailand to the 173 ports worldwide with an annual handling volume of more than 500,000 TEU (20-foot equivalent unit), including empty containers, as in Kosuge et al. [16]. In addition to the top 20 local carriers, 14 local liner shipping carriers are added from MDS Containership Databank [39] to ensure that the coverage of vessel capacity calling at each port in the terrestrial ASEAN region is more than 95%. Regarding the land network, in addition to the missing road link in Myanmar, the inland water transport along the Ayeyawaddy River is included. Moreover, because the zonal subdivision of Myanmar becomes more detailed (on a prefectural basis) as described below, the nodes are set to be more than one in each zone and road links are added. In the simulation, the following effects are varied for each scenario: the cross-border coefficient of the EWC and trucking speed in its Myanmar section; the presence, truck speed and cross-border coefficient of the SC; and the presence of Dawei port and the liner services that call there. The other information on each link remains fixed and unchanged.

The interregional cargo shipping demand (OD matrix) to/from the terrestrial ASEAN countries, (obtained from the World Trade Service (WTS) data by IHS [40]), is divided into zones based on their regional share of the economic index shown in Table 2. Gross regional product (GRP), is used as a regional indicator for dividing the OD matrix for Myanmar. It is estimated by dividing the GDP of the country, by the land cover data for agriculture, and night light data representing manufacturing and service industries, obtained from Kudo and Kumagai [22].

**Table 2.** Zoning method for each terrestrial ASEAN country.

| Country | Zone | Indicator | Source |
|---------|------|-----------|--------|
| Myanmar | 70 | GRP | Kudo and Kumagai [22] |
| Thailand | 77 | GRP | Statistics Ministry of Thailand [41] |
| Vietnam | 62 | Trade volume | Finance Ministry of Vietnam [42] |
| Cambodia | 24 | Sales and GDP growth by region | Kosuge et al. [16] |
| Laos | 17 | GRP | Kudo and Kumagai [22] |

*4.3. Model Calculations*

The GLINS model has a nested structure in which the stochastic network assignment model on the virtual intermodal network is the upper model and the user equilibrium assignment models on the real network in each mode are the lower models. As proposed by Shibasaki [13], the solution to the entire model is obtained by using one set to find the solution to each of the lower-order and upper-order models, and then performing iterative calculations until convergence is reached. As convergence is not guaranteed for the calculation of the entire model, we check it ex-post. However, this still does not guarantee uniqueness of the solution, which is an issue to be addressed in the future.

**5. Model Validation**

In this section, we confirm the reproducibility and validity of the model in terms of container throughput in port and modal shares in the terrestrial ASEAN countries. For the modal share, we focus on the international transport route between Myanmar and Thailand and conduct a sensitivity analysis of the variables included in the cost function.

*5.1. Baseline Scenario Setting and Container Throughput*

Based on the results of our field survey in Myanmar and related literature (Japan Marine Equipment Association [43] and Ministry of Land, Infrastructure, Transport and Tourism (MLIT) [44]), the following scenario is adopted as the baseline scenario for this analysis.

- Railway service: Speed—10 km/h; Frequency—7 trains/week; Handling time—24 h; Distance-proportional cost—1.75 USD/TEU/km,
- Inland waterway transport service: Speed—10 km/h; Frequency—7 services/week; Handling time—48 h; Distance-proportional cost—0.75 USD/TEU/km,
- Level of service in the EWC: Truck speed in Myanmar/Laos/Vietnam section—20 km/h; Thai section—40 to 50 km/h,
- Cross-border coefficient: $\lambda_a = 0.4$,
- The SC and Dawei port: not available,
- Variance parameters for stochastic assignment: $\theta = 0.01$,
- Shipper's time value of freight: $vt = 0.5$ (USD/TEU/hour).

The land cargo flows estimated in the baseline scenario are shown in Figure 3.

Table 3 compares the model-estimated laden container throughputs (excluding transshipment containers) in the ports of the terrestrial ASEAN region with the observed figures of 2016. The maximum error rate between the country-based estimated and observed throughputs is found in Cambodia, which represents the necessity of calibration adopted in Kosuge et al. [16]. Specifically, they calibrated cross-border coefficient $\lambda_a$ based on the interview and field surveys; therefore, the model fitness would be improved if the coefficients were similarly fine-tuned for each terrestrial ASEAN border. The error rates for countries other than Cambodia are only a few percent.

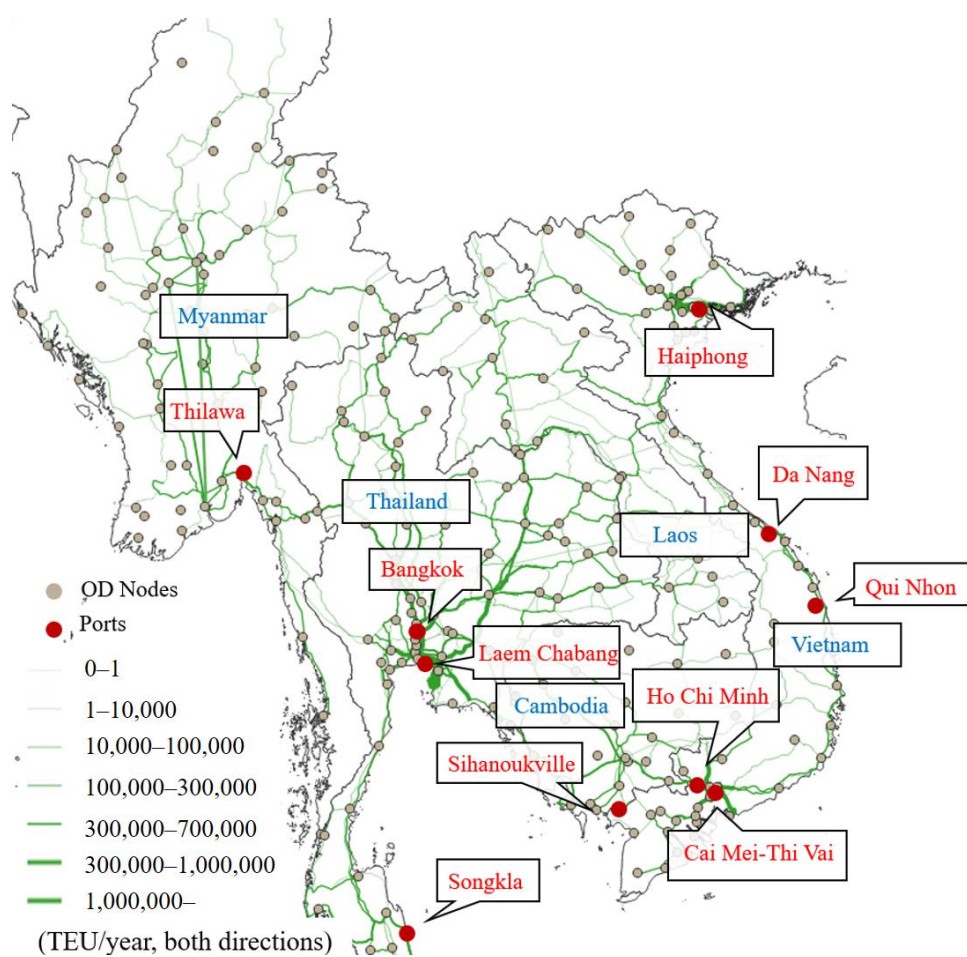

**Figure 3.** Land cargo flows of the baseline scenario estimated in this study (as of 2016).

**Table 3.** Estimated laden container throughput in each port in the terrestrial ASEAN region (baseline scenario, 2016).

| Country | Port | Observed (A) (TEU) | Estimate (B) (TEU) | Difference (A)–(B) (TEU) | Error Rate |
|---|---|---|---|---|---|
| Vietnam | Haiphong | 708,921 | 3,141,070 | 2,432,149 | 304.1% |
| | Da Nang | 233,815 | 141,384 | −92,431 | −39.5% |
| | Qui Nhon | 76,840 | 27,764 | −49,076 | −63.9% |
| | Ho Chi Minh | 4,354,555 | 2,407,315 | −1,947,240 | −44.7% |
| | Cai Mep Thi Vai | 947,317 | 815,257 | −132,060 | −13.9% |
| | Vietnam Total | 6,321,448 | 6,532,790 | 211,342 | 3.3% |
| Cambodia | Sihanoukville | 367,880 | 303,614 | −64,266 | −17.5% |
| | Cambodia Total | 367,880 | 303,614 | −64,266 | 17.5% |
| Thailand | Laem Chabang | 5,105,178 | 5,430,096 | 324,918 | 6.4% |
| | Bangkok | 974,112 | 462,490 | −511,622 | −52.5% |
| | Songkhla | 86,135 | 546,910 | 460,775 | 534.9% |
| | Thailand Total | 6,165,424 | 6,439,496 | 274,072 | 4.4% |
| Myanmar | Thilawa | 319,146 | 333,225 | 14,079 | 4.4% |
| | Myanmar Total | 319,146 | 333,225 | 14,079 | 4.4% |

Further, the largest difference between the port-based estimated and observed throughput is found in Vietnamese ports, including Hai Phong in the north and Ho Chi Minh and

Cai Mep in the south. This is because the value of trade in each province is used as an indicator in the regional division of container shipping demand in Vietnam, as shown in Table 2. More specifically, according to our estimation, the container shipping demand in the Red River Delta and the Northern Priority Economic Region centered on Hanoi, the largest city in the north, would share 32.1% of the total cargo volume in Vietnam in this study whereas that in the Southeast and the Southeast Priority Economic Region centred on Ho Chi Minh City, the largest city in the south, share 45.9%. However, the trade value we adopted in this study includes cargoes other than container cargoes. Among them, air cargo accounts for a large share in value terms; for example, Korean companies have been producing significant quantities of IT-related equipment in and around Hanoi since 2009 which are exported mainly by air. According to Inter National Civil Aviation Organization [45], in Vietnam, the air cargo volume is almost the same at Hanoi airport (314,312 tons, 2016) and Ho Chi Minh City (304,314 tons). Therefore, the actual share of container shipping demand in the southern region, mainly Ho Chi Minh City, would be much larger than that in the northern region, mainly Hanoi, rather than our estimation. In this manner, the maritime container shipping demand in the northern part of the country may be overestimated if the country's container shipping demand is divided according to regional trade value. The improvement on this point is an issue for the future.

Similarly, the estimated throughput in Bangkok port is smaller than the observed figure, whereas the estimated throughputs in the two adjacent ports, Laem Chabang and Songkhla, are larger than observed. This is mainly because the capacity constraint of the port is not taken into account and the calculation of equilibrium assignment does not converge. In particular, the calculation results of container throughputs between Bangkok and Laem Chabang port are heavily fluctuated, because Bangkok port is located nearer to Bangkok, the capital city of Thailand, thus, the hinterland transport cost from it is much cheaper whereas Laem Chabang port provides many liner services with larger containerships resulting in cheaper maritime shipping cost. Incorporating the port capacity constraint and incremental assignment into the model are possible solutions as a further research. Meanwhile, the estimated result in Thilawa, Myanmar, is the same as the country-specific error shown in Table 3, because it is only included as Myanmar's container port in the model.

*5.2. Model Share and Sensitivity Analysis of International Transport between Myanmar and Thailand*

Figure 4 compares the model-estimated modal share of the international transport between Myanmar and Thailand (land transport vs. maritime shipping) with several variations of the cross-border coefficient $\lambda_a$ between Thailand and Myanmar, and the observed ones obtained from the WTS Data [40]. As shown in the figure, if $\lambda_a = 0.4$, the share of land transport is 85.4%, which is closest to the observed share of 83.2%. Further, the figure indicates that if the cross-border coefficient $\lambda_a$ between Thailand and Myanmar increases (i.e., the cost and time of crossing the land border increases), cargo between the two countries shifts from land transport to maritime shipping and the land share decreases.

In summary, although errors at/of container throughput are observed in some ports, the authors consider that the model with the proper cross-border coefficient is confirmed and validated as a whole.

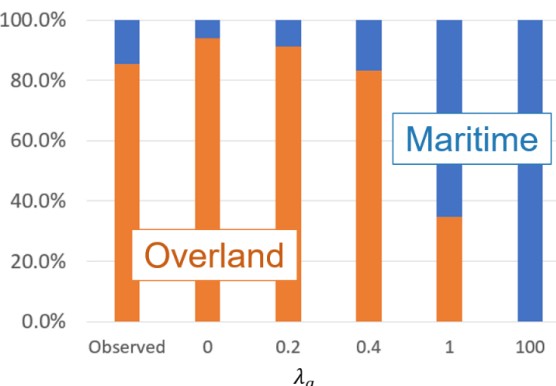

**Figure 4.** Shares of land transport for Myanmar–Thailand transport for each cross-border coefficient and observed values.

## 6. Policy Simulations for GMS Economic Corridors

In this section, the model developed in the previous section is used to analyze the policy scenarios on the GMS economic corridors as follows:

Scenario 1 (S1): Infrastructure development of the EWC.
Scenario 2 (S2): Construction and improvement of the SC and Dawei port.

### 6.1. Infrastructure Development of the EWC

Among the main land transport routes between Myanmar and Thailand, the section between Yangon and Tak in Thailand is duplicated or overlapped with the EWC. However, whereas its Thai section has been improved, the Myanmar section has not yet been fully developed as described in Section 3. In the following scenarios, we assume the transport environment in the Myanmar section of the EWC and border barriers on the EWC are improved. Specifically, (a) the improvement of truck speed in the Myanmar section of the EWC and (b) the simplification of customs procedures on the Myanmar–Thailand border (Myawaddy–Mestho) on the EWC are assumed as shown in Table 4.

**Table 4.** Scenarios set for infrastructure development in the EWC (East–West Corridor).

| Scenario | Truck Speed in Myanmar Section of the EWC (km/h) | Cross-Border Coefficient $\lambda_a$ on Myanmar–Thai Border on the EWC |
|:---:|:---:|:---:|
| Base | 20 | 0.4 |
| S1-1 | 50 | 0.4 |
| S1-2 | 80 | 0.4 |
| S1-3 | 20 | 0 |
| S1-4 | 20 | 0.2 |
| S1-5 | 20 | 0.6 |
| S1-6 | 20 | 1 |

#### 6.1.1. Truck Speed Improvement in the EWC

Regarding the scenarios with varying truck speeds in the EWC (S1-1 and S1-2), Figure 5 shows the estimation results of the cargo volume passing through the EWC at the Myanmar–Thai border (in both directions, the same applies hereinafter unless otherwise noted) and the container throughput of Thilawa port (sum of export and import but only laden containers—the same applies hereinafter unless otherwise noted). The cargo volume passing through the EWC increases by 0.7% (+1087 TEU) in S1-1 and 4.9% (+7514 TEU) in S1-2, compared with the baseline scenario, as truck speeds of the Myanmar section of the

EWC increase. Meanwhile, the container throughput in Thilawa port remains unchanged in S1-1 and decreases by 1.7% ($-5564$ TEU) in S1-2 from the baseline scenario.

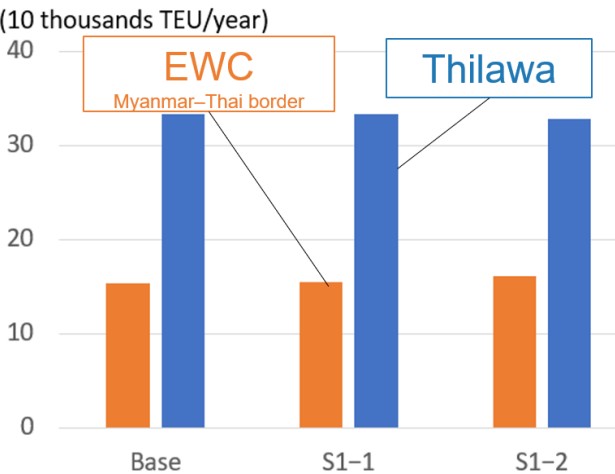

**Figure 5.** EWC transit cargo volumes and container throughput in Thilawa port based on truck speed improvement scenarios.

In summary, as the truck speed of the EWC increases, the cargo volume passing through the EWC increases whereas the container throughput in Thilawa port decreases, but insignificantly.

### 6.1.2. Border Barrier Change in the EWC

Figure 6 shows the estimation results of cargo volume passing through the EWC at the Myanmar–Thai border and the container throughput in Thilawa port for the scenarios on changes in the cross-border coefficient $\lambda_a$ between Myanmar and Thailand on the EWC. Note that the cross-border coefficient on the EWC is changed from the baseline scenario whereas those in other borders are not changed, unlike the sensitivity analysis on the cross-border coefficient shown in Section 5.2.

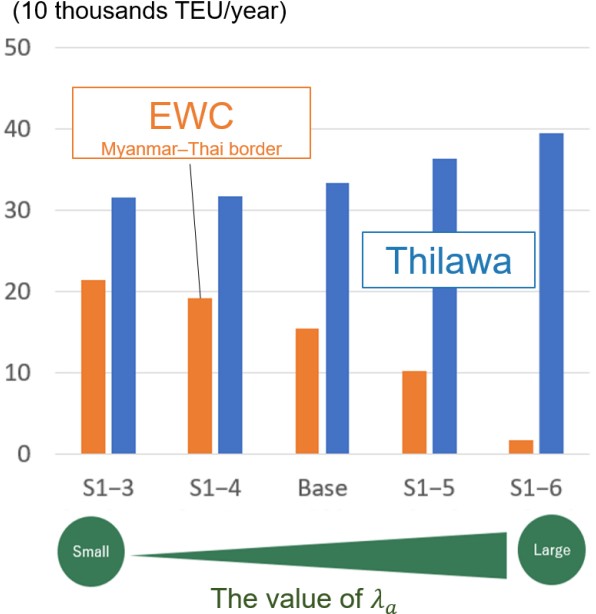

**Figure 6.** EWC transit cargo volumes and container throughput in Thilawa port in the EWC cross-border coefficient change scenario.

Figure 6 reveals that the cargo volume passing through the EWC decreases as the cross-border coefficient on the EWC increases. Meanwhile, the container throughput in Thilawa port increases proportionately as the cross-border coefficient increases; but decreases less with a reduction in the cross-border coefficient. Figure 7 shows the difference in land cargo flows in S1-3, which is the case where the cross-border coefficient $\lambda_a$ is zero, compared with the baseline scenario. As shown in the figure, cargo flow in the EWC at the Myanmar–Thai border in S1-3 increases significantly (59,874 TEU) compared with the baseline scenario and 4037 TEU are shifted from the land border in northern Myanmar.

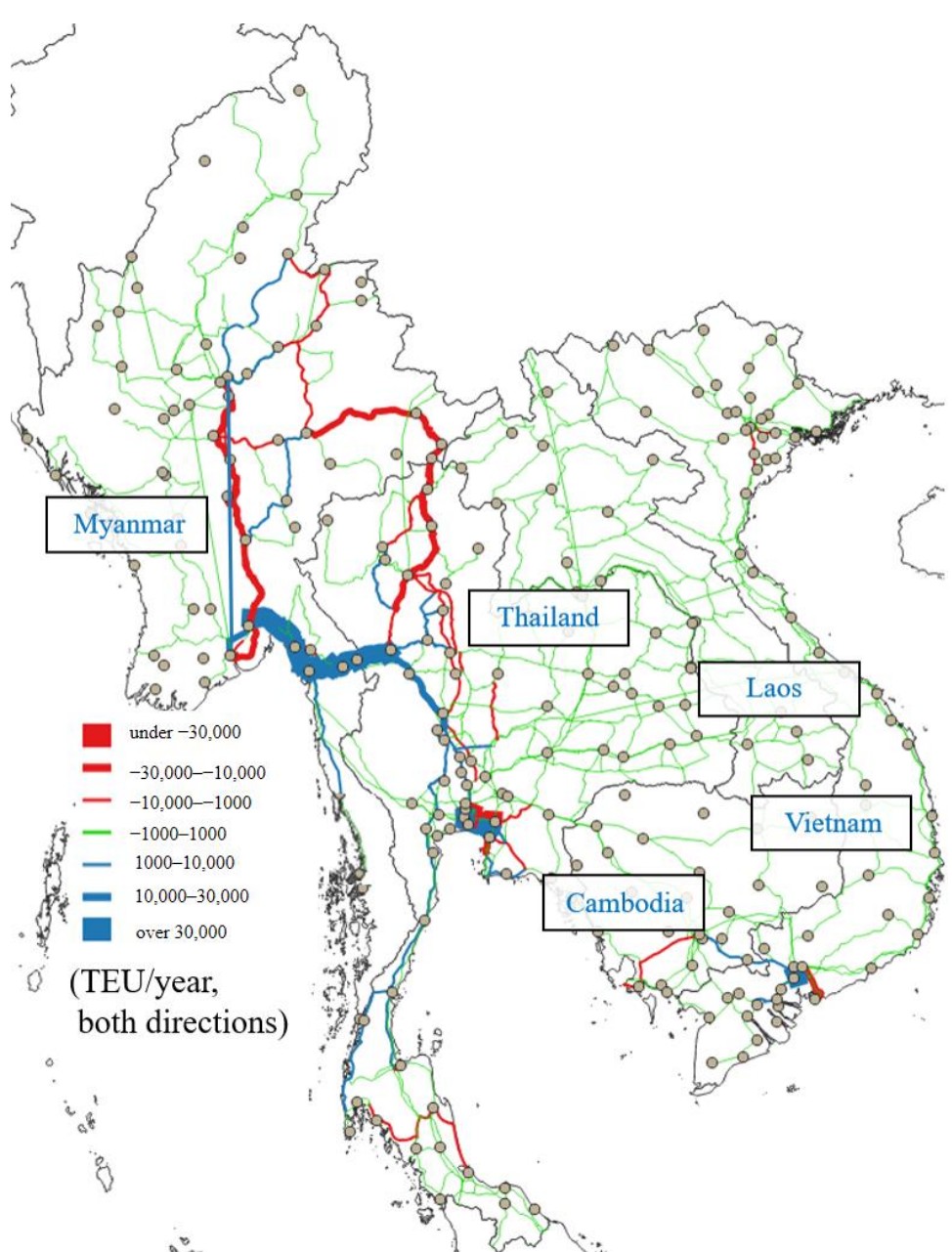

**Figure 7.** Difference in land cargo flows in S1-3 from the baseline scenario.

One of the reasons why the decrease in container throughput in Thilawa port is not large, is that some cargo (10,400 TEU) to and from the regions in Thailand located close to the border with Myanmar, now use Thilawa Port via the EWC instead of Thai ports such as Laem Chabang and Bangkok. Another reason is that the shift from maritime shipping to land transport to and from Thailand weakens the attraction of Thai ports and enhances

that of Thilawa port. The decrease in cargo flow to and from Laem Chabang port can be observed in Figure 7.

Figure 7 also reveals that the improvement of the EWC does not significantly affect countries of terrestrial ASEAN other than Myanmar and Thailand, because the trade volume between Myanmar and these countries is small and more than two international borders have to be crossed if land transport is used. Similar geographical coverage of the affected countries is observed in the other scenarios including the S2 scenarios for the SC and Dawei port.

*6.2. Construction and Improvement of the SC and Dawei Port*

As mentioned in Section 3, the Myanmar section of the SC (between Dawei and Poonamrong) is still undeveloped. Currently, most of the international maritime containers in Myanmar are exported and imported at Yangon or Thilawa port. However, both are river ports with insufficient water depth to accommodate large vessels. Further, these ports are geographically far from the trunk liner service route between East Asia and Europe, which makes it difficult for these ports to attract large vessels. On the other hand, Dawei port in southern Myanmar, has a geographic advantage enabling the development of a deep-water terminal and in being closer to the trunk route than Yangon. Moreover, if the SC becomes available, it will also be closer to Bangkok. From the Thai side, the SC and Dawei port can be positioned as an outer port of Thailand providing a significant shortcut to India, Africa and Europe, avoiding going around the Malay Peninsula and Malacca Strait by vessel. Based on these backgrounds, the impacts of the development of the SC and Dawei port are simulated. Specifically, two policies are envisioned: (a) the development of the SC; and (b) the establishment and increase of liner services calling at Dawei port.

The specific settings of each scenario are shown in Table 5. In S2-1, the link between Dawei and Phu Nam Rong, Thailand, is added as the SC. In S2-2 to S2-4, among 22 liner services that called at Yangon or Thilawa port in 2016, 21 services to/from Southeast Asia and Northeast Asia are assumed to call at Dawei port. The difference between the three scenarios are the timing of port calls: for northbound, southbound and both directions. Further, the truck speed of the SC is changed in S2-5 and S2-6. Moreover, in S2-7, all 14 services connecting Colombo or southern Indian ports (e.g., Chennai) with Southeast Asia or the innermost ports of the Bay of Bengal (i.e., Bangladesh ports and Kolkata/Haldia in India) are assumed to call at Dawei port. Finally, in the last two scenarios, the connection to Europe is considered. In S2-8, the Asia–Mediterranean Sea–East coast of North America service, which returns to Europe from Laem Chabang port, is changed to return from Dawei port. Additionally in S2-9, not just one service that calls at Chennai on the Asia–Europe route, but two services with the largest vessel size on the Asia–Europe route are added (all services are assumed to call at Dawei port only for westbound voyages).

Figure 8 shows the container throughput at Dawei and Thilawa ports and the estimated volume of cargo passing through the EWC and SC at the Myanmar–Thai border in each scenario. Table 6 shows their breakdown by import and export or by direction.

6.2.1. Development of the SC

First, we examine the results of S2-1, which adds the SC to the land transport network, allowing travel at 20 km/h, but does not include the opening of Dawei port. The cargo volume passing through the SC at the Myanmar–Thai border is 66,364 TEU, whereas the cargo volume passing through the EWC at the Myanmar–Thai border decreased by 46,130 TEU, as shown in Figure 8 and Table 6. Hence, the SC becomes a competitor to the EWC for transport between Bangkok and Yangon. However, the total cargo volume passing through the EWC and SC in S2-1 increases by 13% compared to the volume passing through the EWC in the baseline scenario, indicating that these corridors in Myanmar are more frequently used in S2-1 as a whole. Meanwhile, the container throughput in Thilawa port decreases to 329,378 TEU in S2-1 from 333,225 TEU in the baseline scenario; this quantum of decrease is smaller than the quantum increase in the corridors. This may be due to the

opening of the SC which caused the shifting of cargo to land transport via the SC from maritime shipping via Thai ports. This may have resulted in weakening the attraction of Thai ports and expanding the hinterland of Thilawa port. Figure 9 describes the difference in land cargo flow in S2-1 from the baseline scenario and reveals that container flows near Thai ports, north of Bangkok and along the EWC are decreasing, whereas container flows along the SC are increasing.

**Table 5.** Scenario settings for the development of the SC (Southern Corridor) and Dawei port.

| Scenario | Availability of the SC and the Pattern of Calls at Dawei Port | SC Speed (km/h) |
|---|---|---|
| Base | Without the SC and Dawei port | − |
| S2-1 | SC only added | 20 |
| S2-2 | In addition to S2-1, all services calling at Thilawa port call at Dawei port for northbound | 20 |
| S2-3 | In addition to S2-1, all services calling at Thilawa port call at Dawei port for southbound | 20 |
| S2-4 | In addition to S2-1, all services calling at Thilawa port call at Dawei port for both northbound and southbound | 20 |
| S2-5 | Same as S2-4 | 10 |
| S2-6 | Same as S2-4 | 40 |
| S2-7 | In addition to S2-4, 14 new South Asia services call at Dawei port | 20 |
| S2-8 | In addition to S2-7, 1 new European services call at Dawei port | 20 |
| S2-9 | In addition to S2-9, 3 new European services call at Dawei port | 20 |

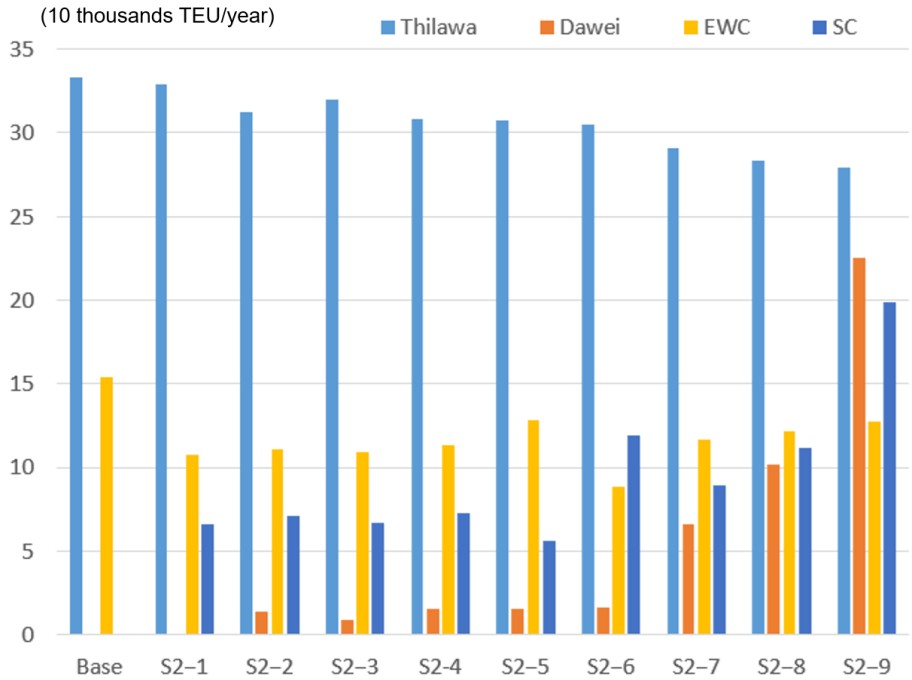

**Figure 8.** Container throughput of Dawei and Thilawa ports and cargo volume passing through the EWC and SC at the Myanmar–Thai border in each scenario.

**Table 6.** Breakdown of container throughput in Thilawa and Dawei ports and the cargo volume passing through the EWC and SC at the Myanmar–Thai border by direction (or by import/export) in each scenario (TEU/year).

| Scenario | Thilawa | | Dawei | | EWC | | SC | |
|---|---|---|---|---|---|---|---|---|
| | Export | Import | Export | Import | Thailand to Myanmar | Myanmar to Thailand | Thailand to Myanmar | Myanmar to Thailand |
| S2-1 | 107,820 | 221,559 | 0 | 0 | 83,747 | 24,008 | 62,832 | 3532 |
| S2-2 | 106,753 | 205,934 | 3490 | 10,085 | 87,314 | 23,823 | 67,338 | 3481 |
| S2-3 | 103,074 | 216,453 | 4318 | 4164 | 85,363 | 25,243 | 64,341 | 3501 |
| S2-4 | 103,584 | 204,381 | 5820 | 9763 | 88,268 | 24,925 | 69,187 | 3463 |
| S2-5 | 103,552 | 204,250 | 5782 | 9785 | 102,785 | 25,872 | 53,740 | 2516 |
| S2-6 | 103,572 | 203,755 | 5784 | 9763 | 71,286 | 24,928 | 87,130 | 3509 |
| S2-7 | 93,582 | 196,868 | 34,248 | 31,724 | 88,918 | 27,391 | 79,720 | 9146 |
| S2-8 | 92,422 | 197,121 | 39,082 | 39,088 | 88,908 | 28,843 | 83,282 | 14,172 |
| S2-9 | 87,884 | 186,456 | 160,734 | 49,732 | 99,420 | 29,034 | 172,304 | 13,377 |

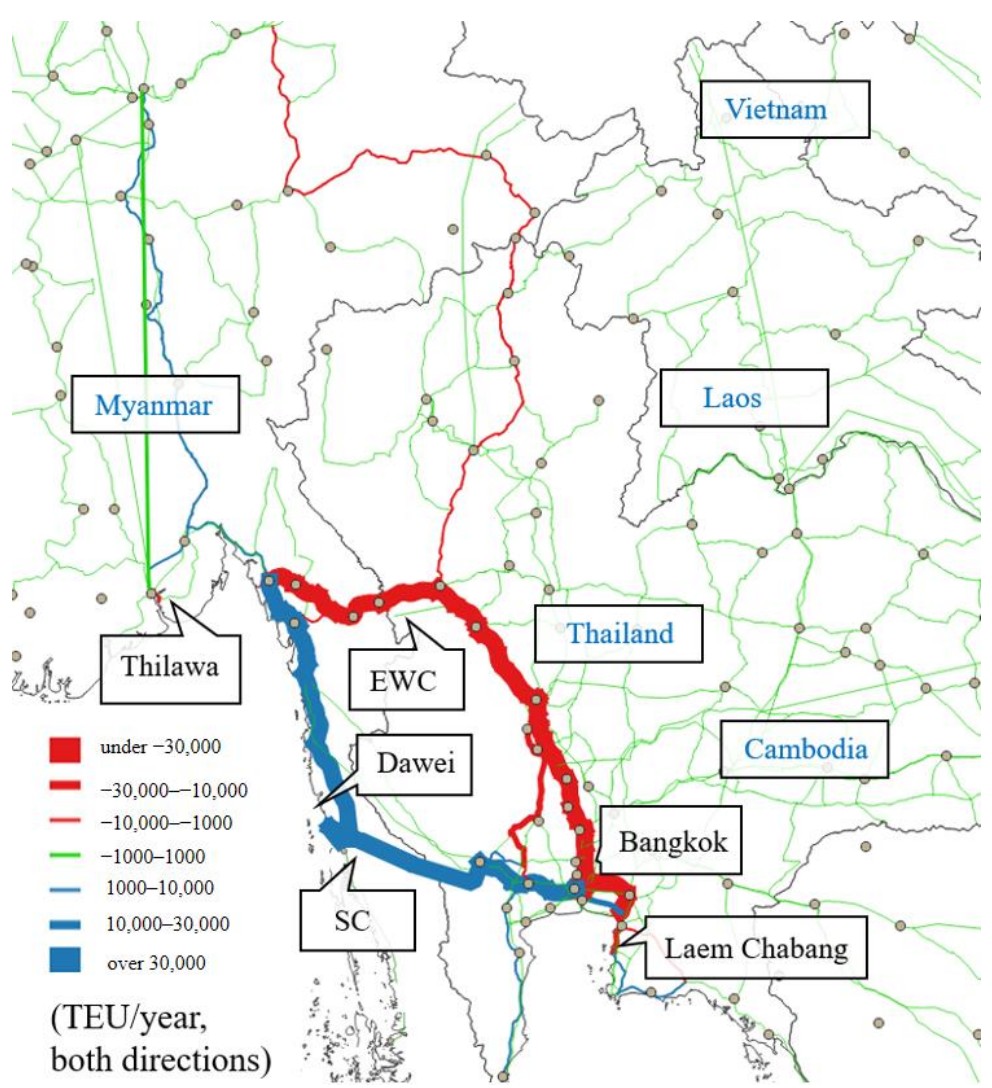

**Figure 9.** Difference in land cargo flows around the SC in S2-1 from the baseline scenario.

### 6.2.2. Opening of Dawei Port and the Calling of Liner Services that Call at Thilawa Port

In S2-2, S2-3 and S2-4, we assume the opening of Dawei port and, that all the liner services calling at Thilawa port will also call at Dawei port, except for one service connecting to Colombo port. In other words, Dawei port is positioned as a feeder port of major Southeast Asian ports such as Singapore and Malaysian ports in these scenarios. As shown in Figure 8 and Table 6, the container throughputs in Dawei port are around 10,000 TEU in these scenarios, which are lower than for Thilawa port. The cargo volumes passing through the EWC and SC at the Myanmar–Thai border increase slightly from S2-1 (up to 4000−5000 TEU). Table 6 reveals that some cargo imported from Malaysia and Singapore shifts to Dawei from Thilawa port in S2-2. This is because the import container volume in Dawei port in S2-2, (where northbound liner services call at Dawei port), is larger than in S2-3, in which southbound liner services call at Dawei port. Regarding Thilawa port, import container volume in S2-2 is smaller than in S2-3. Moreover, most containers exported from Dawei port in S2-2 and imported into Dawei port in S2-3 are considered as domestic transport to and from Thilawa port; in other words, some cargo between Yangon and Thailand via the SC is transported by maritime shipping between Thilawa and Dawei ports. The results in S2-4 have both characteristics of S2-2 and S2-3. In particular, the export container volume from Dawei port as well as the cargo volume from Thailand to Myanmar passing through the EWC and SC are largest among the three scenarios.

In S2-5, in which truck speed in the SC is decreased from S2-4, the cargo volume passing through the SC decreases and that passing through the EWC increases, whereas, in S2-6, where truck speed in the SC is increased from S2-4, the cargo volume passing through the SC increases and that passing through the EWC decreases. There are no significant changes in the container throughput in Thilawa and Dawei ports in these scenarios.

### 6.2.3. Calls of Bay of Bengal Service to Dawei Port

In S2-7, based on the setting in S2-4, 14 trans-Bay of Bengal services are assumed to call at Dawei port, linking southern Indian ports in the Bay of Bengal (e.g., Chennai port) and Colombo port with Southeast Asian ports, or the innermost ports of the Bay of Bengal including Bangladesh's Chittagong port and India's Kolkata and Haldia ports. As shown in Figure 8, the container throughput in Dawei port increases by 50,389 TEU compared to S2-4 and the cargo volume passing through the SC at the Myanmar–Thai border increases by 16,216 TEU. In other words, cargo to and from Thailand is transported to the east coast of India and other areas via Dawei port if direct liner services connect to these ports.

Figure 10 shows the difference in land cargo flows estimated in S2-7 from those in S2-1. From the figure, it is apparent that the cargo flow to/from Thai ports such as Bangkok and Laem Chabang decreases, shifting to the SC, and that some cargo to/from northern Thailand is heading to Dawei port via the EWC, instead of using Thai ports.

### 6.2.4. Calls of European Service to Dawei Port

In addition to the setting in S2-7, we assume that one European service calls at Dawei port in S2-8 and three additional European services call there in S2-9. As shown in Figure 8 and Table 6, the laden container throughput at Dawei port increases by 12,198 TEU in S2-8 from that in S2-7, and further by 132,296 TEU in S2-9. The annual laden container throughput in Dawei port is estimated at 210,466 TEUs in S2-9, which is comparable to that of Thilawa port. The cargo volume passing through the SC at the Myanmar–Thai border, as also shown in Figure 11, increases by 8588 TEU in S2-8 and further by 88,227 TEU in S2-9, indicating that approximately two-thirds of the additional cargo handled at Dawei port is cargo to/from Thailand via the SC. The remaining cargo is shifted from Thilawa port or from Thai ports, coming from northern Thailand via the EWC.

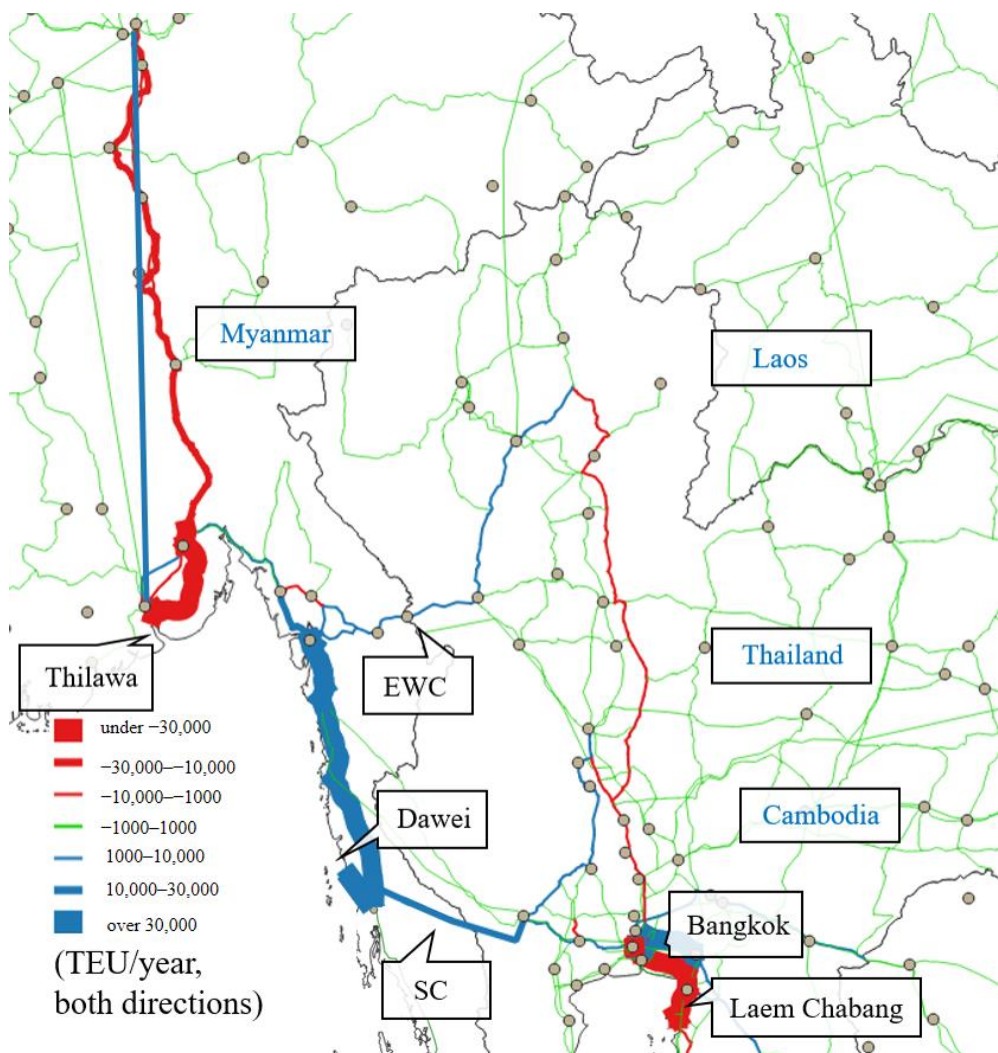

**Figure 10.** Difference in land cargo flows around the SC in S2-7 compared with S2-1.

*6.3. Summary of Policy Simulations*

In the EWC scenarios, the effect of increasing truck speed through road improvements on transport volume was limited, whereas a change in the cross-border coefficient $\lambda_a$ significantly affected transport volume. Specifically, if the cross-border barrier on the EWC is removed (i.e., $\lambda_a = 0$), transit cargo volume would increase by about 40%. Conversely, the volume handled by Thilawa port would not decrease significantly, mainly because cargo to and from the regions in Thailand located close to Myanmar's border shifted to using Thilawa port via the EWC from Thai ports. The shift from maritime shipping to land transport to and from Thailand also weakened the attraction of Thai ports and enhanced the advantages of Thilawa port.

The development of the Myanmar section of the SC encouraged the shift of some portions of cargo, not only from the EWC and Thilawa port, but also from the Thai ports, even though Dawei port was not constructed. Moreover, the Dawei port scenarios showed that the addition of liner services at Dawei port would significantly increase the use of the SC. In these scenarios, significant shifting of cargo from Thai ports to Dawei port was observed, especially in the scenarios where European services were added. Specifically, in S2-9 (which optimistically assumes an increase in port-call services to Dawei port), the volume of cargo handled at Dawei port would increase to 210,466 TEU, whereas the SC transit cargo volume at the Myanmar–Thai border would be 185,681 TEU.

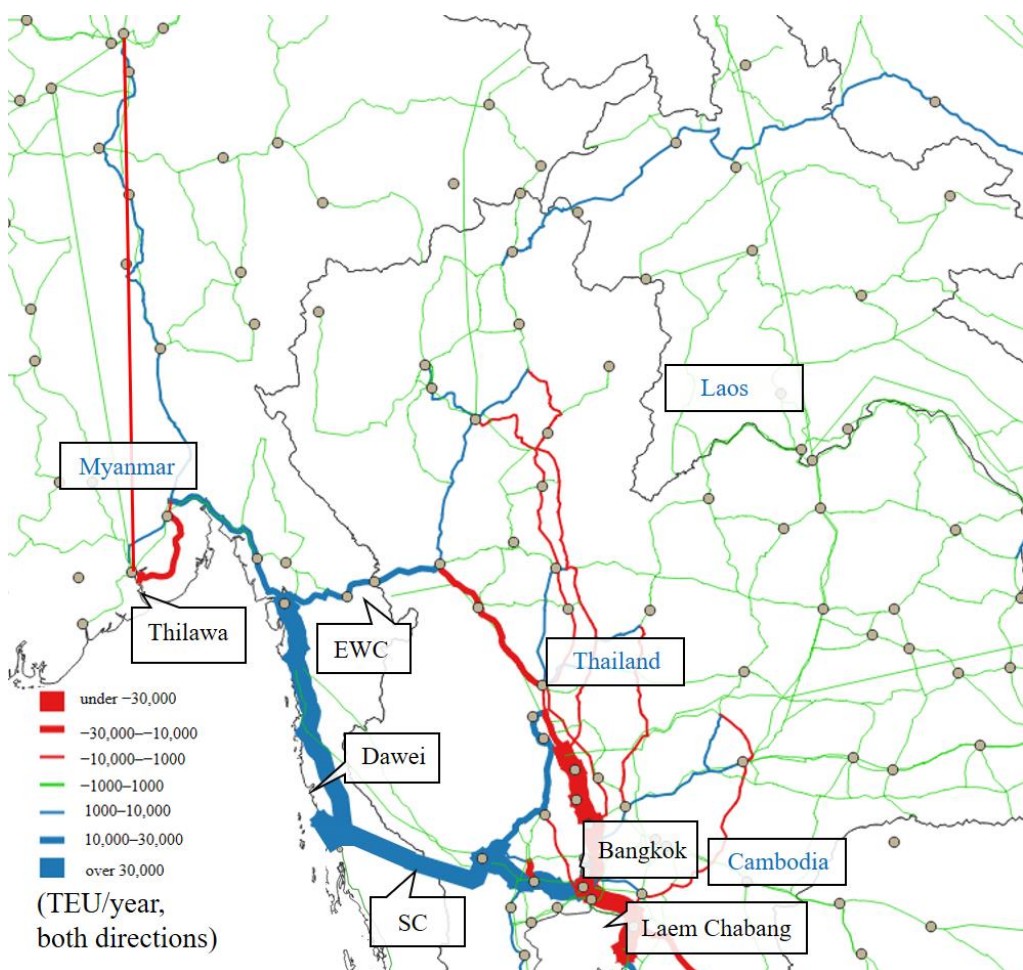

**Figure 11.** Difference in land cargo flows around the SC in S2-9 compared with S2-7.

Regarding the other countries of the terrestrial ASEAN, there was no significant effect of these infrastructural development policies, because their trade volumes with Myanmar are small and more than two international borders have to be crossed if cargo are transported by land.

## 7. Conclusions

In this study, we simulated the international cargo flows in the terrestrial ASEAN region focusing on Myanmar, by using the GLINS model, which was developed by Shibasaki [2,12,13]. Based on the results of the field survey, we updated the input data including detailed zone subdivision and consideration of inland water transport links in Myanmar. We confirmed the validity of the model by comparing the results with observed values of port container throughput and modal share of transport between Myanmar and Thailand, and by conducting a sensitivity analysis to change the cross-border coefficient $\lambda_a$.

Using the developed model, we analyzed policy scenarios for the improvement of the GMS-EWC and the development of the GMS-SC and Dawei port, which are currently planned in Myanmar. Simulations of improvements in truck speed and border barriers in the EWC showed that the improvement in speed has a small effect on the traffic through the EWC but, if the border barrier is reduced, the use of the EWC would increase and the container throughput in Thilawa port would decrease. Simultaneously, as some cargo to and from northern Thailand began to use Thilawa port via the EWC, the reduction in container throughput in Thilawa port would also become relatively low.

The scenarios for SC and Dawei port showed that the development of the SC would not only encourage the shift of cargo from the EWC, but also increase the share of land

transport between Thailand and Myanmar. Furthermore, the scenario for the opening of Dawei port showed that the use of the SC would be expected to increase as the number and variations of liner services calling at Dawei port increase, resulting in a shift of Thai cargo to Dawei port. The significant increase in container throughput in Dawei port was deemed comparable to that of Thilawa port, if the services to connect to eastern India and Europe were added. Thus, unlike the previous models by the authors [14,15], we can simulate individual policies such as the development of the EWC, SC and Dawei port, and obtain reasonable results. Some findings of this study reinforce the implications obtained from previous studies that analyzed individual policies in Myanmar. The results in this study indicated that the combination of opening a new port and a transport corridor would give a more significant and wider impact on cargo flows even for a neighboring country (Thailand), as with Black and Kyu [23] and Isono and Kumagai [27]. This study also revealed that the development of a new port and transport corridors may reduce the congestion of Thilawa port, as Zin [24] pointed out on the dry port in Myanmar.

Meanwhile, there are still several issues to be addressed. First, the validity of the model should be further enhanced. For instance, the calibrations on cross-border coefficient at each national border and consideration of air cargo in the process to make the OD matrix are necessary. As regards to Thailand, model accuracy may be affected by the fact that Laem Chabang and Bangkok ports, which are of different sizes, are located close to each other; therefore, we can consider applying other methods of network assignment. Moreover, the model could be applied to various other policy simulations. For instance, as Nam and Win [25] pointed out, domestic intermodal hinterland transport network including rail and inland water transport should be focused on in further studies. Moreover, although this study focused on the relationship with Thailand, the simulation on the connection with Chinese land networks is also necessary, because Myanmar has a large volume of trade with China and China is also interested in Myanmar to connect with by land for promotion of the Belt and Road Initiative. Further, especially in developing countries, infrastructure investment should be planned based on the expected future economic growth of the country concerned; therefore, the simulations taking into account the future economic growth of terrestrial ASEAN are necessary such as Isono and Kumagai [27]. Furthermore, as mentioned at the beginning of this paper, environmentally sustainable infrastructure development is an essential issue currently. Thus, it is also important to discuss the simulation results of this study from an environmental aspect, especially by quantifying the environmental impact caused by the development of the GMS economic corridor and new ports, as indicated in Sukdanont et al. [26] and Comi et al. [46].

**Author Contributions:** Conceptualization, R.S. and H.S.; methodology, T.Y. and R.S.; software, T.Y.; validation, T.Y. and R.S.; formal analysis, T.Y.; investigation, T.Y., R.S., H.S. and H.U.; resources, R.S. and H.U.; data curation, T.Y.; writing—original draft preparation, T.Y. and R.S.; writing—review and editing, R.S., H.S. and H.U.; visualization, T.Y.; supervision, R.S.; project administration, R.S.; funding acquisition, H.S. All authors have read and agreed to the published version of the manuscript.

**Funding:** This research received no external funding.

**Data Availability Statement:** Restrictions apply to the availability of these data. All data used in this paper were processed by the authors based on the data obtained from the third parties that the authors listed in the references as well as the authors' past studies. Therefore, they are available from the authors with the permission of these third parties.

**Conflicts of Interest:** The authors declare no conflict of interest.

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
