# Peer review of "Impact on Myanmar’s Logistics Flow of the East–West and Southern Corridor Development of the Greater Mekong Subregion—A Global Logistics Intermodal Network Simulation"

_sustainability, doi:10.3390/su13020668_

Round 1
Reviewer 1 Report
The paper presents the results of the GLINS model for Myanmar region for different scenarios. The topic of the paper is interesting, but the research novelty should be firmly emphasized.
The summary of the literature refers to different studies regarding simulation models developed for the ASEAN region. It is important to highlight the peculiarities of the research for the study area, but I recommend a broader review related to simulation models for regional or international logistic systems.
The study aims to identify recommended solutions for the upgrading of the transport infrastructure in Myanmar. The title of the paper refers to logistic efficiency, but the simulation results present only flows assigned to different components of the transport system, without any statements about costs, capacity use, etc. Clearer criteria should be defined for assessment and comparison of the simulation results.
Different scenarios are defined based on synthetic attributes of the transport system of Myanmar. The GLINS model is adapted and applied to obtain the overall flow pattern. Therefore, the calibration and validation steps need stronger argumentation (even if some explanations are given for the differences between the observed and estimated values). I suggest presenting also relative values for the differences between the observed and estimated values (in Table 3).
Supplementary minor comments:
- The quality of Figure 1 should be improved.
- In my opinion, Figures 2, 3 are not appropriate for a scientific paper.
- Several tables need explicit measures and units in columns.
Reviewer 2 Report
Dear Authors,
I think your paper is very interesting. You provide a clear introduction and an adequate justification about why you focus on the analysis and simulation of logistics policies in Myanmar. You also explain that there are no previous studies taking this specific perspective, which is a good point per se.
I overall like the paper, however I have two main concerns that I would define as major, although I am confident that you are able to adress them:
The first concern regards your discussion of the findings. Despite you claim that no previous research takes exactly the perspective of this paper, your model seems to largely rely on a previous publication by the same authors. I think it is very important that you adequately explain what are the differences between the two models and whether the results of this paper are comparable to previous results. More generally, I would like you to compare your findings with those of the similar papers existing in literature and cited in the introduction so as to show how your new research adresses previous gaps and enriches the related literature.
The second concern regards the focus of the paper. While I think the topic is interesting and maritime and inland shipments do have an environmental impact, such considerations are missing in the paper. Given that Sustainability is a journal that pays large attention to environmental issues, I think you should discuss environmental implications of your different findings, scenarios and policies as well (even if in a more qualitative way).
Thanks for your work so far and good luck with your future research in this field
Round 2
Reviewer 1 Report
The new version increases the paper's quality. But several minor adjustments are recommended.
1. The Introduction section becomes very long. I suggest splitting the section into two parts: one introductory with a presentation of the paper objectives and the structure of the paper, and a second one with the presentation of particularities of studies on logistic system of the analysis topic.
2. The coefficient λ (Lambda) is an important parameter in the simulation scenarios. So, a clearer definition for this parameter is required (furthermore, defining a calculus equation for this parameter would help to better understand its significance).
3. The overall quality of the figures should be enhanced. More attention should be paid to the significance and units of the measures represented in graphs. E.g. in figure 5: the cargo volumes are per year? for EWC, the values are averages per one way, per both ways, on the entire corridor or only on the Myanmar section? The same observations apply to the other figures.
Author Response
Thank you for valuable comments again.
1. We divided the introduction section into two sections ("Introduction" and "Literature Review").
2. We added the equation and description related to λ (the notation has been changed to λa) in section 4.
3. We revised figures as well as related sentences in the manuscript to clarify that the results are described in both directions at the national border between Myanmar and Thailand in principle.
Reviewer 2 Report
Thank you for this revision. I find the paper improved.
Author Response
Thank you very much for your comment.